# The Experience of Using Video Support in Ambulance Care: An Interview Study with Physicians in the Role of Regional Medical Support

**DOI:** 10.3390/healthcare8020106

**Published:** 2020-04-23

**Authors:** Veronica Vicente, Anders Johansson, Bodil Ivarsson, Lizbet Todorova, Sebastian Möller

**Affiliations:** 1The ambulance medical service (AISAB), 12118 Stockholm, Sweden; 2Academic EMS, 12118 Stockholm, Sweden; 3Karolinska Institute, Department of Clinical Science and Education at Södersjukhuset, 11861 Stockholm, Sweden; 4Office of Medical Services; Region Skåne, 20525 Malmö, Sweden; anders.johansson@med.lu.se (A.J.); bodil.ivarsson@med.lu.se (B.I.); Lizbet.Todorova@skane.se (L.T.); Sebastian.Strunk-Moller@skane.se (S.M.); 5Department of Clinical Science, Lund University, 22185 Region Skane, Sweden; 6Department of Cardiothoracic surgery, IKVL, Lund University, 22185 Lund, Sweden

**Keywords:** communication, telemedicine, prehospital care, ambulance care, medical support, decision making, triage

## Abstract

*Background:* In order to facilitate more effective patient assessment and diagnostic support by improving the flow of information between ambulance nurses (AN) and physicians in the role of regional medical support (RMS), an application was developed for transmitting real-time video images. *Objective:* The objective of this study was to elucidate the physicians’ experiences using a video application to support the assessment and triage procedure in ambulance care, when patients are deemed to not have an urgent need for emergency care. *Design:* The design for this research was a qualitative interview study. Ten physicians, working as RMS in ambulance care, were purposively selected to participate. The telemedicine concept studied consisted of a real-time video image application, in addition to the currently used mobile phone. When a patient was deemed eligible for inclusion in the study, the ambulance nurse (AN) contacted the RMS via telephone to initiate a video consultation. To elucidate the RMS experience of using the application, a conventional content analysis was performed. *Results:* The main theme “a feeling of being satisfied through a sense of increased patient safety” emerged from the following two categories: adds value in diagnosing situations (three subcategories, i.e., support in diagnosing, usability, and technical weakness) and increase communication opportunities (four subcategories, i.e., assessing the level of care, patient dialogue, professional communication, and team learning). *Conclusions:* Physicians in the role of RMS experienced a positive impact using video image transmission in addition to the currently used mobile phone. This evaluation was derived from a sense of increased patient safety in the assessment situation when patients were considered to be triaged to self-care.

## 1. Introduction

Ambulance services have an important safety-creating function in society and for individual patients, and therefore well-functioning emergency care processes are deemed crucial. It has been previously discussed that all medical activities require competencies based on knowledge and clinical skills, and also need to manage a huge amount of information related to patient care where the primary goal is to provide information to experts when and where it is needed [1].

In recent decades, ambulance care has evolved from a sole transport organization with limited medical skills and few treatment options to an organization with high medical skills and access to a range of different treatment options with high complexity, often fully integrated with hospital-based emergency care [2]. The required professional competence of the ambulance health care staff varies internationally. In the UK, Australia, USA, Norway, and several other countries, the paramedic system is used but, in Sweden and the Netherlands, the registered nurse (RN) is considered to be the most appropriate profession to operate in ambulance care services. Since 2005, the Swedish National Board of Health and Welfare stated that it was mandatory to staff ambulances with RNs (i.e., ambulance nurses, AN). The evolution of the ambulance services has also implied that well-functioning teamwork between AN and other professions (e.g., physicians) in the prehospital space is necessary to meet multifaceted care needs. These needs developed from increasing knowledge about the positive effects of rapid care efforts, more accurate triage, early treatment, and safe transport of patients to the appropriate level of care. As such, teamwork is an important part of being able to provide safe and good care. In addition, the demand for emergency medical services worldwide is predicted to continue to increase [3,4]. The use of technology to accelerate the identification of the patient’s need for medical care, while maintaining quality, is therefore necessary. Valenzuela Espinoza et al. reported that live and bidirectional audio-video telemedicine in stroke ambulance care, is a promising new method to accelerate and improve the quality of emergency stroke care, and therefore even has the potential to be cost-effective in the future [5].

In Skåne County, the southern part of Sweden with 1.3 million inhabitants (population density 123/km^2^), approximately 160,000 ambulance care assessments are conducted annually [6]. In 2017, 69% of the assessed patients were treated and transported to the appropriate care instance, while 31% were assessed and left at home or the place of incidence with advice on self-care or further triaged to primary care. In the case of triage to a different level of care other than emergency departments, the AN must call a regional medical support (RMS) line [7]. This consultation is mandatory, for example, in cases where the RN does not see a need for emergency care or admission to the hospital, or alternatively where patients do not want to be admitted, despite the RNs judgement. The RMS is available to the ambulance staff 24-7, for discussion, support, and decisions in all clinical situations, such as triage, drug prescribing, patient positioning, stay at home, contact in accordance with the care program, or when the teams want support in other areas. Consultations with the RMS typically concern elderly multi-sick patients rather than the most acute patients, since the triage system Rapid Emergency Triage and Treatment System (RETTS) in our region detects about 30% overtriaging by the emergency call centers.

In order to improve the flow of information between AN and the RMS, an application has been developed that allows the transmission of video image from the scene, supporting more effective patient assessment and diagnosis together with the ambulance staff. A recently published study concluded that telemedical concepts in emergency medical services (EMS) lead to improved process times and patient outcomes [8]. The authors also discussed that emergency mobile telemedicine systems are now sufficiently reliable, which should enable widespread use in the future.

While the introduction of digital healthcare has created an opportunity for the physical patient to become digital, a prerequisite for innovative products to become successful is that healthcare providers choose to adopt these novel concepts [9]. To increase the understanding of obstacles to the implementation of concepts of telemedicine, the purpose of this study was to elucidate the physician’s experience of using a video application to support on-scene assessment and triage of patients without an apparent urgent need for emergency care.

## 2. Materials and Methods

This study used a qualitative interview approach. Collected data were analyzed through conventional content analysis following the procedure described by Hsieh and Shannon [10]. In this conventional content analysis, coding categories are derived directly from the text data followed by interpretation of the underlying context.

The study was approved by the Regional Ethical Board in Stockholm, Sweden (Dnr 2017/2547-31/5, 2018). The informants were informed that their responses would be kept confidential and that they could withdraw from the study at any time with no explanation. Written informed consent was obtained from all participants.

### 2.1. Study Setting and Population

The study was carried out in Skåne County in Sweden during 2017. Physicians working as RMS within ambulance care that had used the telemedicine system during the technical evaluation part of the project were eligible for inclusion in the study. Informants were purposively selected to participate in the study, to provide a broad perspective of the use of the system. A total of ten physicians (6 males and 4 females) were included (Table 1).

### 2.2. Study Method

The study evaluated the concept of a real-time video image telemedicine application which aimed to offer more efficient assessment and diagnostic support for discussion and decision in clinical situations where RMS support was needed. In brief, the system consisted of an application on Android-based handheld devices, which were connected via a server-based system. After establishing contact, video signals were transmitted peer-to-peer using encrypted WebRTC streams carried by commercial 3G and 4G networks. The system was to be used after the ambulance team first performed an assessment following current local guidelines. When deemed eligible for inclusion in the technical evaluation, the patient´s consent was obtained and the AN contacted the physician via telephone to initiate a video consultation with the RMS.

### 2.3. Data Collection and Processing

The semi-structured interviews were conducted individually. Each interview was recorded with a digital recorder and started with a short presentation of the study’s aim. The total interview time with all informants was 5 h and 51 min, with a min-max interview time ranging from 19 to 42 min. The opening question asked the RMS to describe experiences regarding the use of telemedicine, when AN wished to refer the patient to self-care or if the patient was considered to be triaged to another care level. The second question focused on whether the RMS had experienced any added value for the patient.

### 2.4. Sample Size

Acording to the aim of the present study we included all 10 RMS that had used the system in Skåne County (Sweden) during the technical evaluation. According to Kvale and Brinkman [11] this sample size is considered sufficient for this type of study.

### 2.5. Data Analysis

All interviews were transcribed verbatim by a medical secretary. Then, the interviews were condensed, abstracted, and grouped into subcategories, categories, and one main theme. To ensure that the classification was as free from bias as possible, could be replicated, and was reliable, all authors discussed the subcategories and categories and, afterwards, adjustments were made until consensus was reached. An example of the analysis process is presented in Table 1.

## 3. Results

Informant characteristics are presented in Table 2. The main theme “A feeling of being satisfied through a sense of increased patient safety” emerged from the following two categories (and seven subcategories): (1) adds value in diagnosing situations (support in diagnosing, usability and technical weakness) and (2) increase communication opportunities (assessing the level of care, patient dialogue, professional communication, and team learning) (Table 3).

### 3.1. Adds Value in Diagnosing Situations

#### Support in Diagnosing

The RMS described how the concept added value in diagnosing situations. The RMS were able to inspect the patient’s body language, skinsuit, and breathing. Moreover, RMS were able to guide and ask questions to the AN in order to complete further investigations such as palpation or neurological status while watching the patient. As a result, the diagnosis could be made faster and the concept was experienced by RMS as an increase in patient safety.

### 3.2. Usability

The RMS argued that the concept had the necessary technical capabilities and found that the concept was both easy to learn and easy to use. However, there were also concerns that the total duration times for all ambulance missions could increase and the RMS commented that there was a risk that both AN, patients, and relatives through the use of the system could experience a loss of integrity, for example, when they had been awakened in the middle of the night for an RMS consultation and all participants saw each other less “freshly”.

### 3.3. Technical Weaknesses

The RMS argued that the device was slow to get to work adequately, and thus caused unnecessary delays. In addition, the RMS complained that the extra device was easy to forget or break and perceived weaknesses in relation to sound quality. In terms of image quality, there were split experiences and image quality problems appeared usually to be associated with the small size of the device. A short battery life and long charging time were also described as a problem. Furthermore, the RMS also wished for a device where they could have access to the patient’s hospital records.

### 3.4. Increase Communication Opportunities

#### Assessing the Level of Care

The RMS observed that ambulance staff sometime asked the RMS to talk directly to patients or relatives during the assessment when, in the RN’s opinion, the patient was in need of admission to the hospital. The RMS were convinced that the concept was a good complement to a telephone consultation in these situations, since they could see the patient and, in some cases, appear on the video for patients and close relatives. They expressed that the video image helped to triage to the appropriate level of care, including giving advice on self-care; however, the RMS still expressed concern regarding missing necessary investigations such as X-ray and lab tests when leaving patients at home.

Another concern with the concept that the RMS raised was that they did not want to be seen as “gatekeepers” to the healthcare facilities. They also saw a risk that some patients would take the opportunity to get a “physician consultation” free of charge, while the RMS group was solely intended for the ambulance services and should not be seen as a medical visit on the internet.

### 3.5. Patient Dialogue

For ambulance staff, it is important to gain patient confidence and the RMS pointed out that the AN have a major responsibility to introduce the RMS if the support is to be used. The RMS expressed that the video image helped to increase patient safety, participation, and trust by facilitating an understanding that the ambulance staff works together with a physician for confirmation or counseling. However, the RMS noted that certain patients do not want to feel exposed for cultural reasons, and therefore the ambulance staff should be prepared to alert those patients.

### 3.6. Professional Communication

An advantage of the concept that was mentioned was that the number of communication opportunities between the AN and RMS increased by looking at each other, thus strengthening teamwork. Conversely, concerns were raised that trust could be lost if the ambulance staff felt controlled and the RMS even worried that ambulance staff would feel insulted if the RMS took over too much of the contact with the patient.

### 3.7. Team Learning

For the RMS and the ambulance staff, the profession involves continuous learning and some RMS noted that the concept could be of value in mutual learning situations. In the case of troublesome patient cases, the AN and RMS could discuss different solutions more easily. Furthermore, AN perceived that they were able to better express their thoughts, while the RMS provided instruction on the use of different methods of examination and followed the outcome through the video. In contrast, the RMS indicated that there was no time for active learning in the patient situations, and therefore learning should take place in other situations.

## 4. Discussion

With the introduction of more powerful commercial 3G/4G networks that allow the transmission of large amounts of data, the use of video-based telemedicine has become feasible and has evolved greatly into the current generation of systems that are under evaluation in several countries and for a number of medical conditions, such as stroke [8,12].

Although telemedicine in many of these studies has been applied to support RMS in more acute situations as compared with our study, the implementation of these systems face similar challenges. A recurrent theme in these studies is technical feasibility, with available bandwidth as one of the main factors. With the introduction of the more powerful commercial 3G /4G networks, it seems that the challenges faced in earlier studies have been overcome. One study [4] addressed this question specifically, reporting connectivity data from a densely populated in the Brussels area. These data resemble recent data obtained in a partly rural landscape in southern Sweden [13].

While using only a single 3G/4G connection, the current study suggests that the RMS, in ambulance care, experience a feeling of being satisfied with the technical concept studied and a sense of increased patient safety in the assessment situation. The technical concept was designed to conduct a digital video transmission in the prehospital space. Since the technical solution did not involve any storage of images or video, the concept does not conflict with applicable laws on information security. During development, the concept was improved during an iterative process and tested before use in the study. The final acceptance of the system by participants was ensured so that the concept worked properly and there were no further adjustments needed before being used in clinical situations. It could be argued that a larger pilot study should have been conducted prior to use in the study, for example, to address the identified noise and image issues. Notwithstanding these minor issues, the interviewed RMS agreed that the concept has development potential.

The RMS accepted the concept as a complement to normal telephone consultation in situations of support and triage since they could see the patient and sometimes close relatives. The RMS also expressed that the video image facilitated the triage to the appropriate level of care, including giving advice in self-care situations. This result is in line with a previous study that showed that the use of high-quality telemedicine in stroke cases was feasible and had a positive impact on local stroke care [14]. Our study was not designed to investigate the possible impact of the use of telemedicine on medical outcomes, for example, whether or not more accurate triage occurred. However, we believe our study suggests indirectly that the combination of AN’s information and the RMS visual assessment can contribute to a better outcome of the triage situation.

Manojlovich and DeCicco [15] pointed out the importance of effective communication between nurses and physicians to reduce the risk of mistakes in health care. An advantage with the present concept that was acknowledged by the RMS was that communication opportunities increased between the AN and RMS. By being able to look at each other, and sometimes the patient, the teamwork seemed strengthened through improved communication possibilities. The RMS sense of improved communication is supported by a study by Fairbanks et al. [16], which found that the majority of communication events between nurses and physicians in an emergency department was during visual contact (face-to-face) rather than via phone. In line with these findings and our own study, we believe that visual communication via video streaming contributes to the RMS sense of improved patient safety through fewer mistakes and misunderstandings.

Beyond this perceived sense of improved patient safety, the RMS expressed that the video image helped to increase patient participation and trust by conveying an understanding that the ambulance staff works together with a physician, ensuring sound decisions and triaging. Earlier studies in nurse telephone counseling demonstrated that some patients felt the need to fight to be respected and to avoid being rejected [17]. Using video streaming in ambulance care could result in a decreased number of emergency ward visits in proportion to the number of emergency calls, with more satisfied patients since the patients could take part in the physicians’ advice. However, further studies are needed to investigate these aspects more completely.

The RMS also expressed that they did not want to be “gatekeepers” to the healthcare facilities and saw a risk that some patients would take the opportunity to get a “physician consultation” free of charge. This concern is similar to findings by Rosén et al. [18]. In focus group interviews with nurses in ambulance care, informants hinted that some patients could have another agenda when calling the ambulance service, such as that some patients see the ambulance services as a mobile care facility for obtaining an immediate assessment. If this phenomenon among patients is common or becomes more common, we consider this a concern important enough to warrant further investigations, since resources in ambulance healthcare are not unlimited. Possible risks with this behavior include reduced access to ambulance-based emergency services even for patients with higher priority. The Swedish Health Care Act states that “care must be of good quality and meet the patient’s individual needs”. Accordingly, the Swedish health care services aim to strengthen the patients’ involvement in their own care, indicating a potential situation of conflict. Therefore, ensuring patient safety is a priority both in administrative and organizational terms to resolve this situation, emphasizing the importance of effective communication between physicians, nurses, and patients in ambulance care. Further studies are, therefore, necessary to assess how the technology can be improved, as well as how the organization, community costs, and above all, the patients’ health are affected.

## 5. Limitations

The selected method has obviously affected the study in certain ways, and therefore it is important to put findings in context of the conduct of the study. Our data analysis followed the established qualitative content analysis process by Hsieh and Shannon [10] and we consider that the credibility of our results is strengthened by the agreement of all authors on the results. Different researchers’ preunderstanding of an area or phenomenon can be of great importance in all parts of the research process [19]. The researchers’ standpoint could be affected by multiple factors, such as culture, educational level, personal believes, or professional practice experiences. To actively raise different researcher awareness bias due to certain preunderstanding could be discouraged, but probably although not fully avoided. The authors of this research have had reoccurring discussions during the research process and the authors’ backgrounds as ambulance nurses and university teachers, therefore, have similar contextual backgrounds. However, the risks of investigator bias increases with knowledge and presumptions of an area, especially during interpretation of qualitative data, but preunderstanding can also be considered to be an asset in order to produce initiated research questions or define implications of research. This article is a product of consensus between the present authors, and therefore is believed to have raised awareness and reduced investigator bias, which we believe enhances the conformability. However, despite methodology considerations, the results can only be assessed by the external reader who is familiar with the context.

## 6. Conclusions

Physicians in the role of regional medical support experienced a positive effect using video support to complement consulting assignments in ambulance care through a sense of increased patient safety in the assessment situation, when patients were treated at the scene or considered triaged to self-care. However, the used concept should be further developed since the RMS experienced minor image and audio weaknesses. Hopefully, the results of this study should make the available knowledge in the field more nuanced and raise awareness of the value of diagnostic support for AN and the RMS in order to optimize patient care and treatment in the prehospital environment.

## 7. Article Summary

### 7.1. Why Is This Topic Important?

Voice only callbacks have the potential for introducing patient safety issues into triage situations in ambulance care. The use of telemedicine for AN–RMS communication has not yet been studied from an experience point of view.

### 7.2. What Does This Study Attempt to Show?

This study elucidates physicians´ experiences of using a video application to support the assessment procedure in ambulance care when patients are considered not to have an urgent need for emergency care. This study aims to describe the RMS experiences regarding the use of VIA, and whether the RMS experienced any added value for the patient.

### 7.3. What Are the Key Findings?

The RMS experienced a positive impact of using video to complement consulting assignments in ambulance care, through a sense of increased patient safety in the assessment situation.

### 7.4. How Is Patient Care Impacted?

This study does not directly evaluate how patient care is impacted. However, our study suggests that the use of telemedicine increases patient safety through improved assessment. Further studies are needed to investigate patient outcomes in more detail.

## Figures and Tables

**Table 1 healthcare-08-00106-t001:** Example of the analysis process, from verbatim meaning unit to the main theme.

Meaning Unit	Code	Subcategory	Category	Main Theme
"*Generally, I experience the device as a contribution. I could see the patient and can even ask individual questions. I get an idea of the patient’s general condition and I could hear how the person communicates with me...or I could see how the person moved* ".	Complements phone with visual impression	Support in diagnosing	Adds value in diagnosing situations	A feeling of being satisfied through a sense of increased patient safety
"*On the other hand we were also very cautious that this support system should not be a way if you [patients] only wishes to visit a doctor*”.	Risk for misuse of image communication	Assessing the level of care	Increases communication opportunities	

**Table 2 healthcare-08-00106-t002:** Informant characteristics. Gender and years of experience as physician in the role of regional medical support (RMS) are presented as absolute (*n*), min-max, and relative frequencies (%). Frequencies of use of the image/video system/informant (System use) are presented as mean and min-max values.

Values	Male	Female	Total
*n*/%	6/60	4/40	10/100
Years as RMS			
mean	7	4	6
(Min-max)	(1–10)	(0.5–5)	(0.5–10)
System use			
(*n*)	4	5	4
(Min-max)	(2–10)	(1–12)	(1–12)

**Table 3 healthcare-08-00106-t003:** Summary of the subcategories, categories, and the main theme.

Subcategories/ Selected Informants Quotes	Categories	Main Theme
Support in diagnosing	*Adds value in diagnosing situations*	*A feeling of being satisfied through a sense of increased patient safety.*
*"Generally, I experience the device as an added value. I could see the patient and could even ask individual questions. I got an idea of the patient’s general condition and I could hear how the person communicated with me...or I could see how the person moved ".* (RMS 4)		
Usability		
*"Should video consultation be introduced to all [patients], the total care time will increase ... and the ambulance staff already have so much equipment and now they will carry an additional device”.* (RMS 8)		
Technical weaknesses		
*"The batteries did not properly work and it was just so uncomfortable that we lost some flexibility".* (RMS 9)		
Assessing the level of care	*Increases communication opportunities*	
*"Do you talk about abdominal patients ... if you think of telephone conversations ... now I instead had a video consultation where I saw it was a huge patient so it is not possible to palpate…it means very much, so for me, it feels like patient safety enhancing because you get a lot of attention in the picture".* (RMS 2)		
*"On the other hand, we were also very cautious that this support system should not be a way if you (patients) only wish to visit a doctor”.* (RMS 1)		
Patient dialogue		
*"It is very important how AN communicate with the patient” e.g." We will make a video consultation”…and if you are not prepared for it as a patient, you find yourself in a difficult situation. Better is if the AN in some professional way, prepared the patients e.g…”my doctor is able to see you for a second assessment…is that ok? That’s why professionalism is important to the patient”.* (RMS 6)		
Professional communication		
*"If you can increase the communication situation and you act as a team around the patients, even if you’re not geographically in the same place ... I’m convinced that teamwork increases… video consultation should be seen as something positive and not as something difficult or controlling…I can see positive effects for all collaborators".* (RMS 4)		
Team learning		
*“I think you can, in some way, find yourself in a learning situation. You maybe do not learn more specific medical facts, but you learn to understand. The relationship between AN and RMS is very important, to be able to consult each other in a better way with the video… It adds a dimension to the phone”.* (RMS 6)		

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
