# Peer review of "The Experience of Using Video Support in Ambulance Care: An Interview Study with Physicians in the Role of Regional Medical Support"

_healthcare, 2020, doi:10.3390/healthcare8020106_

Round 1

Reviewer 1 Report

This paper proposes to give support to physicians by using a
video application for assessment and triage procedure in ambulance care.

The manuscript is well written, however, it is possible to find some grammatical errors. Therefore, a careful review is needed.

The introduction is very brief and out of date, requiring stronger articles from the current state of the art to improve the study background. Here are my suggestions:

"A comprehensive review on smart decision support systems for health care." IEEE Systems Journal 13.3 (2019): 3536-3545. DOI: 10.1109/JSYST.2018.2890121

"Time gain needed for in-ambulance telemedicine: cost-utility model." JMIR mHealth and uHealth, v. 5, n. 11, p. e175, 2017. DOI: 10.2196/mhealth.8288

Figure 1 is not present in the manuscript (line 135, p. 3). Table 2 needs to be adjusted, the reader can easily fail to understand it, as well as Table 2.

Include suggestions for future work.

References are poorly formatted and in some cases, information is missing. Besides, they do not follow a defined pattern. It is unacceptable for the authors to neglect this part.

Author Response

Response to Reviewer 1 Comments

The introduction is very brief and out of date, requiring stronger articles from the current state of the art to improve the study background. Here are my suggestions:

"A comprehensive review on smart decision support systems for health care." IEEE Systems Journal 13.3 (2019): 3536-3545. DOI: 10.1109/JSYST.2018.2890121

"Time gain needed for in-ambulance telemedicine: cost-utility model." JMIR mHealth and uHealth, v. 5, n. 11, p. e175, 2017. DOI: 10.2196/mhealth.8288

Response 1: The section Introduction is complemented with recommended references.

Figure 1 is not present in the manuscript (line 135, p. 3). Table 2 needs to be adjusted, the reader can easily fail to understand it, as well as Table 2.

Response 2:  Figure 1 is mentioned in the text under section Data analysis and Table 2 is adjusted, making it hopefully more understandable.

Include suggestions for future work.

Response 3: Suggestions according future work are included in last sentence in section Discussion.

References are poorly formatted and in some cases, information is missing. Besides, they do not follow a defined pattern. It is unacceptable for the authors to neglect this part.

Response 4: References are formatted throughout the manuscript.

Reviewer 2 Report

This is an interesting study with potentially important implications.

I would suggest in the introduction discussing potentially what are the implications of having a more efficient healthcare system. I am not familiar with the Swedish system, but in the United States, efficiency is always something that healthcare administrators and managers strive for given the amount of scarce resources (let alone during this current epidemic). Also, the introduction alludes to it but it should be made more explicit that emergency departments are unlikely to have the capacity to  accommodate the 160,000 patients assessed by the ambulatory staff.

Line 48-51. The second half of the sentence is awkwardly written.

Line 62-63. I am not sure that multi-sick is grammatically correct. Most acute patients… this sounds incomplete.

Font is inconsistent at times, like at line 167.

Author Response

I would suggest in the introduction discussing potentially what are the implications of having a more efficient healthcare system. I am not familiar with the Swedish system, but in the United States, efficiency is always something that healthcare administrators and managers strive for given the amount of scarce resources (let alone during this current epidemic). Also, the introduction alludes to it but it should be made more explicit that emergency departments are unlikely to have the capacity to accommodate the 160,000 patients assessed by the ambulatory staff.

Response 5: We discuss in section Discussion that using video streaming in ambulance care may result in a smaller number of emergency ward visits in proportion to the number of emergency calls, but that we also suggests that further studies have to investigate these aspects more properly.

Line 48-51. The second half of the sentence is awkwardly written.

Response 6: The sentence has been revised.

Line 62-63. I am not sure that multi-sick is grammatically correct. Most acute patients… this sounds incomplete.

Response 7: The section has been revised to improve clarity

Font is inconsistent at times, like at line 167.

Response 8: Mentioned Font-issues are corrected.

Reviewer 3 Report

General Comments:

Thank you for the opportunity to review this manuscript. Firstly, I would like to express my general impression of the manuscript. The study topic is of great interest for the development of distance assessment of patients and for ambulance care. I understand that “editorial stuff” can happen when text is entered into a manuscript template. However, the text style differ for example fonts and text size and some parts are missing. Therefore, this manuscript does not feel properly prepared before being sent to this journal. Simultaneously, this is something that can be rapidly addressed when the text is revised. I have also some specific comments to do.

Line 7 Author List

…Bodil…. RN…The surname is missing.

Line 35 Keyword

You use the keyword “prehospital care”. However, in the article you use ambulance care. Why isn't ambulance care used as a keyword?

Introduction

Line 40-45

The text describes the development of ambulance care, which seems to be mainly from a Swedish context. Description of the development can be interesting. However, has nothing happened at all since 2005 (the reference provided is old)? To understand the context better, please describe how ambulance care it is today and if the development in Sweden might differ from other countries/continents.  

Line 46-47

To understand the context better, please describe the different professions more (for example formal competencies). What is an ambulance nurse? What are the differences between an AN and a registered nurse (RN)? What kind of physicians are referred? A specialist in ambulance care?

Line 48-49

What kind of demand is expected to increase? Please, clarify what is meant.

Line 57

In your text, it seems that triage is used to decide the place where care and treatment is provided. Is this triage? It looks like a tool for logistics rather than a tool to decide the order (how long the patient can wait) of care and treatment. Please, clarify what is meant.

Line 62-63

The text is somewhat unclear. Do you mean that multi-patients are not acute patients? Please, clarify what is meant.

Line 75

What kind of patients are included in this group? Elderly multi-sick or what? Please, clarify what you mean with urgent need.

Materials and Methods

According to the “Instructions for Authors” and “Manuscript Preparation” the next heading after the introduction is “Materials and Methods”. Please, adjust the heading according to the instructions.

Study design and ethical considerations

The font size differs from the other headings

Line 80

You claim that a conventional content analysis has been done according to Hsieh & Shannon. However, in the result section (line 146-147) you inform that you also will present the quantity of statements in your subcategories. For me it is unclear how you have done your data analysis. Is it a qualitative or quantitative content analysis? Is it really a conventional content analysis and not some sort of summative content analysis? Please, clarify your method so it will match the way you present your results.

Table 1

Table 1 can be made somewhat clearer. Under "Years as RMS" you indicate that it is mean, but not under "System use".

Line 112–115

Measure of interest is a repetition of the purpose and do not adds something new.

Data Collection and Processing

Your purpose was “to elucidate the physicians’ view of using…” However, the interview questions were about RMS experiences of the use and value for the patients. “View of using” may as well be about attitudes rather than experiences of something. Therefore, it is somewhat unclear how your purpose of the study matches your interview questions. Please, clarify your interview questions related to your purpose.

Sample size

You included 10 RMS and according to Kvale & Bringman would it be enough for this type of study. However, to evaluate the sample size, important information about the data collection is missing. Please, clarify how your interviews were in terms of min-max time and total interview time.

Additional comments

Please, describe:

1) the authors experience and pre-understanding of ambulance care and this specific topic.

2) whom of the authors that transcribed the interviews and whether it was verbatim or not.

Results

The result consists of 7 subcategories, 2 categories and 1 main theme. However, the main theme has not been included in the result.

Please, explain how subcategories became categories. Guess you've done some sort of abstraction, although I don't see this in the result.

Please explain why it is important to know how many statements each sub-category has. Don't really see the relevance of this.

Quantitative data in the result is rarely interesting when it is a qualitative study unless the choice of the method says otherwise. Examples: some RMS, most RMS etc.

Table 2

I think you could report the results in different ways. However, quotation should support the content. As the table is now designed, the citations seem to support the subcategory headings more. This makes the table difficult to understand. Suggests another design if a table is to be used.

Limitations

Line 291

Given that Healthcare is a journal which publishes work in the interdisciplinary area of all aspects of medicine and health care research. Is your choice of journal appropriate if the results can only be assessed by readers who know the context? Is it not your responsibility to describe the context as carefully as possible to facilitate the readers who have limited knowledge of the context? I can agree that it is the reader who have the responsibility to determine the transferability of the results to another context or EMS outside Skane county or Sweden.

Author Response

The text style differ for example fonts and text size and some parts are missing. Therefore, this manuscript does not feel properly prepared before being sent to this journal. Simultaneously, this is something that can be rapidly addressed when the text is revised.

Response 9: We apologize, apparently something went wrong during the submission procedure ... but now the fonts should be correct.

Line 7 Author List

…Bodil…. RN…The surname is missing.

Response 10: Surname included.

Line 35 Keyword

You use the keyword “prehospital care”. However, in the article you use ambulance care. Why isn't ambulance care used as a keyword?

Response 11: In Sweden, ambulance care is often also referred to as pre-hospital care. However, in Sweden, pre-hospital care can also include primary care and other "out-of-hospital" facilities. But we have complemented our keyword list with ambulance care.

Introduction

Line 40-45

The text describes the development of ambulance care, which seems to be mainly from a Swedish context. Description of the development can be interesting. However, has nothing happened at all since 2005 (the reference provided is old)? To understand the context better, please describe how ambulance care it is today and if the development in Sweden might differ from other countries/continents.  

Response 12: We have now extended the section Introduction with more detailed information with regard to ambulance care development in Sweden.

Line 46-47

To understand the context better, please describe the different professions more (for example formal competencies). What is an ambulance nurse? What are the differences between an AN and a registered nurse (RN)? What kind of physicians are referred? A specialist in ambulance care?

Response 13: Please see answer Response 12.

Line 48-49

What kind of demand is expected to increase? Please, clarify what is meant.

Response 14: We have now clarified what was meant.

Line 57

In your text, it seems that triage is used to decide the place where care and treatment is provided. Is this triage? It looks like a tool for logistics rather than a tool to decide the order (how long the patient can wait) of care and treatment. Please, clarify what is meant.

Response 15: We have further explained the role of RMS in Swedish ambulance care.

Line 62-63

The text is somewhat unclear. Do you mean that multi-patients are not acute patients? Please, clarify what is meant.

Response 16: We have now clarified in more detail why not all patients are acute patients.

Line 75

What kind of patients are included in this group? Elderly multi-sick or what? Please, clarify what you mean with urgent need.

Response 17: Here we would like to refer to our response 16, which clarifies that all patients receiving an ambulance, are not in need of immediate emergency medical care as our triage system results in a significant over triage.

Materials and Methods

According to the “Instructions for Authors” and “Manuscript Preparation” the next heading after the introduction is “Materials and Methods”. Please, adjust the heading according to the instructions.

Response 18: Headings throughout the manuscript are adjusted.

Study design and ethical considerations

The font size differs from the other headings

Response 19:
Headings are adjusted.

Line 80

You claim that a conventional content analysis has been done according to Hsieh & Shannon. However, in the result section (line 146-147) you inform that you also will present the quantity of statements in your subcategories. For me it is unclear how you have done your data analysis. Is it a qualitative or quantitative content analysis? Is it really a conventional content analysis and not some sort of summative content analysis? Please, clarify your method so it will match the way you present your results.

Response 20: Thank you for pointing this out. We have done a conventional content analysis, but sometimes (other manuscripts) we have got requests from reviewers to sum-up the number of statements that can be attributed to analysed subcategories and categories. However, we agree that this type of “quantifying results” may feel confusing, and in line with your wishes, we have removed all summative results according to statements’ in the text and in all tables and figures.

Table 1

Table 1 can be made somewhat clearer. Under "Years as RMS" you indicate that it is mean, but not under "System use".

Response 21: Now clarified in legend to Table 1.

Line 112–115

Measure of interest is a repetition of the purpose and do not adds something new.

Response 22: We agree and have deleted the section.

Data Collection and Processing

Your purpose was “to elucidate the physicians’ view of using…” However, the interview questions were about RMS experiences of the use and value for the patients. “View of using” may as well be about attitudes rather than experiences of something. Therefore, it is somewhat unclear how your purpose of the study matches your interview questions. Please, clarify your interview questions related to your purpose.

Response 23: Thanks for pointing this out. We agree that this might be inconsistent use of language; we have therefore replaced “views” with “experiences”, in line with our intention.

Sample size

You included 10 RMS and according to Kvale & Bringman would it be enough for this type of study. However, to evaluate the sample size, important information about the data collection is missing. Please, clarify how your interviews were in terms of min-max time and total interview time.

Response 24: Requested interview times are now presented under the section Data Collection and Processing.

Additional comments

Please, describe:

1) the authors experience and pre-understanding of ambulance care and this specific topic.

2) whom of the authors that transcribed the interviews and whether it was verbatim or not.

Response 25: 1. That pre-understanding is an important factor in qualitative analysis is described in more detail in section Limitations.…and 2, the “transcribe-process” is described under section Data analysis.

Results

The result consists of 7 subcategories, 2 categories and 1 main theme. However, the main theme has not been included in the result.

Response 26: The main theme is included in the result.

Please, explain how subcategories became categories. Guess you've done some sort of abstraction, although I don't see this in the result.

Response 27: As described under heading Data analysis, the interviews were condensed, abstracted and grouped into sub-categories, categories and to one main theme. To ensure that the classification was as free from bias as possible, all authors discussed the sub-categories and categories until consensus was reached.

Please explain why it is important to know how many statements each sub-category has. Don't really see the relevance of this.

Response 28: Please see response 20.

Quantitative data in the result is rarely interesting when it is a qualitative study unless the choice of the method says otherwise. Examples: some RMS, most RMS etc.

Response 29: In line with your earlier recommendations we have excluded “qualitative” words e.g. some/most etc.

Table 2

I think you could report the results in different ways. However, quotation should support the content. As the table is now designed, the citations seem to support the subcategory headings more. This makes the table difficult to understand. Suggests another design if a table is to be used.

Response 30: We have deleted all quotations in the text and re-designed the tables, hopefully in a more stringent way.

Limitations

Line 291

Given that Healthcare is a journal which publishes work in the interdisciplinary area of all aspects of medicine and health care research. Is your choice of journal appropriate if the results can only be assessed by readers who know the context? Is it not your responsibility to describe the context as carefully as possible to facilitate the readers who have limited knowledge of the context? I can agree that it is the reader who have the responsibility to determine the transferability of the results to another context or EMS outside Skane county or Sweden.

Response 31: Since we now have, in several different ways, described the context in more detail, described how an ambulance organization can be organized and how, in general, what healthcare personnel can experience using innovative telemedicine, we think our results can be of value in other parts of health care where an efficiency improvement involve the increased collaboration between professionals’.

We thank the reviewers again for their valuable comments and we hope that our changes are in line with their criticism. Thank you!

Round 2

Reviewer 1 Report

The authors addressed all my suggestions. I consider the manuscript OK for publication.

Author Response

Thank you for your great inputs in this manuscript. 

Reviewer 3 Report

Thank you for the opportunity to re-review this manuscript.  I can see that you have re-written the manuscript and made it clearer. However, I have two comment:

Line 55-56

Your reply is OK for me. However, I will remember that you in Sweden have a specialist training for nurses in ambulance care at master level. Is it not true that an AN is such a specialist trained nurse?

Results

I understand your analysis process and the various steps you have taken. However, it is still unclear how your subcategories develop the categories and how the categories develop your main theme. What characterizes, for example, "Adds value in diagnosing situations" and what makes "Adds value in diagnosing situations" and "Increase communication opportunities” to become " A feeling of being satisfied through a sense of increased patient safety”? The headings should be based on the content that could be understood as something, this something needs to be described. For the moment, your categories and main theme seem to be more “headlines” without content.

Author Response

Response 1: The National Board of Health and Welfare in Sweden controls the Swedish ambulance services. In addition to the general regulation that at least one registered nurse (RN) should staff each Swedish ambulance, the different health care region or county council has the authority to choose their own rules according to skills and staffing. Therefore, some regions have decided that the RNs should also hold a specialist degree in nursing (master degree) in order to work in the ambulance services. Therefore, the responsible nurse in each ambulance can be a RN, with or without a specialist education at the master's level, but with various specialties e.g., ambulance-, anesthesia-, intensive- or primary care. But still, all categories of nurses working in the context are called an ambulance nurse (AN).

Response 2: We believe, that based on our original descriptions of the manifest content (meaning units and codes), i.e. near the text, subcategories and categories are interpreted (emerged) as a latent part of the total content, perhaps sometimes far from the text but still close to the participants' experiences. This underlying interpretation then becomes the underlying meaning or "red thread" in the text.

Closeness to the text therefore means concrete descriptions (meaning units and codes) and the sub-categories and categories can therefore be seen as an interpretation, which meant that we sorted the content into our categories and then continued to search for the common latent content and formulated this as a theme. Therefore, during the analysis process, the authors have taken different scientific positions linked to the study's goals and therefore we consider that the theme can be a reasonable interpretation from the categories, as described in table 2.

We also believe that in qualitative content analysis you can choose whether you want low or high degree of abstraction in headings. We have chosen present levels but, we have re-written the text at the beginning of the results section, which we think is more in line with the abstract.

We thank the reviewer again for the valuable comments. Thank you!